# Juvenile Hake *Merluccius gayi* Spatiotemporal Expansion and Adult-Juvenile Relationships in Chile

Daniela V. Yepsen [1], Luis A. Cubillos [1,2,*] and Hugo Arancibia [1]

1 Programa de Doctorado en Ciencias mención Manejo en Recursos Acuáticos Renovables, Facultad de Ciencias Naturales y Oceanográficas, Universidad de Concepción, Concepción 4070386, Chile; daniela.yepsen@gmail.com (D.V.Y.); harancib@udec.cl (H.A.)

2 Centro COPAS COASTAL, Departamento de Oceanografía, Facultad de Ciencias Naturales y Oceanográficas, Universidad de Concepción, Casilla 160-C, Concepción 4070386, Chile

* Correspondence: lucubillos@udec.cl

**Abstract:** The abundance of juvenile fish changes due to endogenous processes, and determining the functional relationships among conspecifics is essential for fisheries' management. The hake (*Merluccius gayi*) is an overexploited demersal fish widely distributed in Chile, from 23°39′ S to 47°00′ S in shallow and deep water over the continental shelf and shelf break. We studied the spatiotemporal distribution of hake juveniles (from ages 0 and 1), emphasizing endogenous relationships among juveniles and adults. The abundance per age data were obtained from bottom trawl cruises carried out in the austral winter between 1997 and 2018. Generalized additive models showed a similar spatiotemporal pattern for ages between 0 and 1, and negative effects of adult hake aged seven and older on the abundance of the young generation. Regarding the changes in juvenile abundance, the residual deviance of selected models explained 75.9% (for the age 0) and 95.3% (for the age 1) of the null deviance, revealing a significant increase in juvenile abundance from 2002 to 2007 and subsequent abundance stability at higher levels. Furthermore, the expansion in the abundance of juveniles after 2002 was favored by the low abundance of older adult hake, most which are able to cannibalize young hake. Our results highlight the importance of endogenous factors in the spatial distribution of Chilean hake juveniles to identify nurseries or juvenile areas free of potential cannibal adults.

**Keywords:** Chilean hake; spatial distribution; GAM; spatiotemporal; endogenous effects





## 1. Introduction

It is critical to understand the causes of commercially exploited fish population distribution [1,2], which could change due to density-independent and density-dependent processes [3,4]. Density-dependent changes are related to changes in predation intensity [5,6], food availability [7,8], or variation in habitat temperature [9,10]. There is also a consensus that density dependence is a feature of population dynamics for most species [11,12]. However, most models of exploited population dynamics assume that density-dependent regulation only affects early life processes [13]. For example, Ohlberger et al. [4] found that the juvenile life stage of Atlantic cod (*Gadus morhua*) is compensatory, and that adult cod cannibalism affects the survival of age-0 cod. Andersen et al. [14] showed that habitat size determines density-dependent regulation and can occur early in large habitats. Consequently, the fishing yield is higher when mainly juvenile fish are exploited as density-dependent regulation occurs at late ages, while adults' exploitation may maximize yield when density-dependent regulation occurs early and through a compensatory stock–recruitment relationship. Lorenzen and Camp [15] provide empirical evidence for determining an appropriate recruitment size or age when juveniles are not subject to density-dependent mortality. In the Northwestern Mediterranean Sea, adults were generally more densely concentrated than juveniles, and occupied areas were included in the distribution of juveniles [16]. In

Gadiformes, juvenile individuals are usually separated from adult fish [17–19]. Understanding the spatiotemporal distribution of juveniles can contribute to identifying nursery areas that could be protected to enhance the recruitment and recovery of fish populations [20].

We evaluated the spatiotemporal effects on the distribution of juveniles and the endogenous effects on the abundance of juveniles from older adults of a cannibalistic species of wide spatial distribution, such as the hake (*Merluccius gayi*) in Chile. The distribution of the species extends along the coast from 23°39′ S to 47°00′ S, with demersal habits at depths between 10 and 500 m [21–23], and greater abundance is found between 31 °S and 41 °S. The acoustic biomass of hake showed an abrupt decrease in 2004, from about 1 million tons before 2003 to 300–400 thousand tons after 2004 [22,23]. At the same time, with the decrease in total abundance, a juvenilization of age composition and decreased maturity length has been observed [24,25].

Reduction in the adult fraction of the stock would result from intense fishing [26], predation by the jumbo squid *Dosidicus gigas* [27], less cannibalism of juveniles [24], and environmental effects [28,29]. However, endogenous relationships between juveniles and adults have not yet been evaluated, mainly regarding whether that relationship is significant and determinant of hake juveniles' temporal and spatial distribution. Indeed, hake is a cannibal, such as other species in the Merluccius genus [30–32], and the effects of density-dependent interactions may inhibit or expand the spatial distribution of juveniles. In this paper, the objective was to determine the spatial distribution of juvenile abundance (between ages 0 and 1) and the endogenous relationship between juvenile and adult hake in a spatiotemporal context.

## 2. Materials and Methods

### 2.1. Study Area and Data

The study area corresponds to the spatial extent of the Chilean hake stock's assessment cruises, carried out from 1997 to 2018 between 29°39′ S and 42°10′ S every year at the same time (Figure 1). The stock assessment cruises are carried out yearly during the austral winter by a staff member of the Instituto de Fomento Pesquero (IFOP), although no assessment cruises were undertaken in 1998 and 2003 [23]. Although the survey is designed for acoustic estimates of biomass, we utilized logbook data consisting of survey catches and fishing effort, length–frequency data, and length–age keys by sex (Table 1). The code of each research survey allowed us to obtain the database with the grant number allocated by Fondo de Investigación Pesquera y Acuicultura (https://www.subpesca.cl/fipa accessed on 7 April 2022) and IFOP (https://www.ifop.cl accessed on 7 April 2022). IFOP provides information of the swept area (km$^2$) for each fishing haul every year.

**Table 1.** Year, survey code, and number of specimens and fishing tows utilized to build the database. Code = grant number of the survey. ALK = age–length key, where n is the number of specimens (=otoliths) utilized to build the ALK for each year. WLD = weight–length data, where n is the total number of specimens utilized to obtain length–weight relationships, and hence the mean weight of length classes. LFD = length–frequency data, where n is the total number of specimens sampled in each year. Tows = number of trawls, where n is the number per year. Source: technical reports available in https://www.subpesca.cl/fipa/ (accessed on 7 April 2022) and https://www.ifop.cl (accessed on 7 April 2022).

| Year | Code | ALK (n) | WLD (n) | LFD (n) | Tows (n) |
|------|--------|---------|---------|---------|----------|
| 1997 | 1997-12 | 972 | 3754 | 23,497 | 133 |
| 1999 | 1999-04 | 999 | 3699 | 15,035 | 135 |
| 2000 | 2000-04 | 1011 | 2216 | 21,952 | 124 |
| 2001 | 2001-18 | 1045 | 2693 | 26,427 | 141 |
| 2002 | 2002-03 | 1138 | 3778 | 29,210 | 153 |
| 2004 | 2004-09 | 1013 | 3624 | 17,570 | 137 |
| 2005 | 2005-05 | 726 | 3321 | 16,516 | 138 |

**Table 1.** *Cont.*

| Year | Code | ALK (n) | WLD (n) | LFD (n) | Tows (n) |
|------|------|---------|---------|---------|----------|
| 2006 | 2006-03 | 1117 | 3599 | 17,819 | 134 |
| 2007 | 2007-16 | 997 | 4669 | 28,080 | 171 |
| 2008 | 2008-14 | 993 | 4240 | 26,648 | 153 |
| 2009 | 2009-13 | 572 | 4335 | 16,673 | 149 |
| 2010 | 2010-10 | 544 | 3226 | 12,102 | 125 |
| 2011 | 2011-03 | 647 | 3800 | 12,310 | 138 |
| 2012 | 2012-04 | 659 | 3648 | 11,375 | 138 |
| 2013 | 2013-12 | 635 | 3679 | 13,645 | 146 |
| 2014 | 682-020 | 627 | 3387 | 13,397 | 136 |
| 2015 | 682-032 | 621 | 2995 | 9419 | 104 |
| 2016 | 682-042 | 653 | 4155 | 13,914 | 145 |
| 2017 | 682-046 | 644 | 3548 | 11,947 | 127 |
| 2018 | 682-056 | 625 | 3813 | 11,935 | 135 |

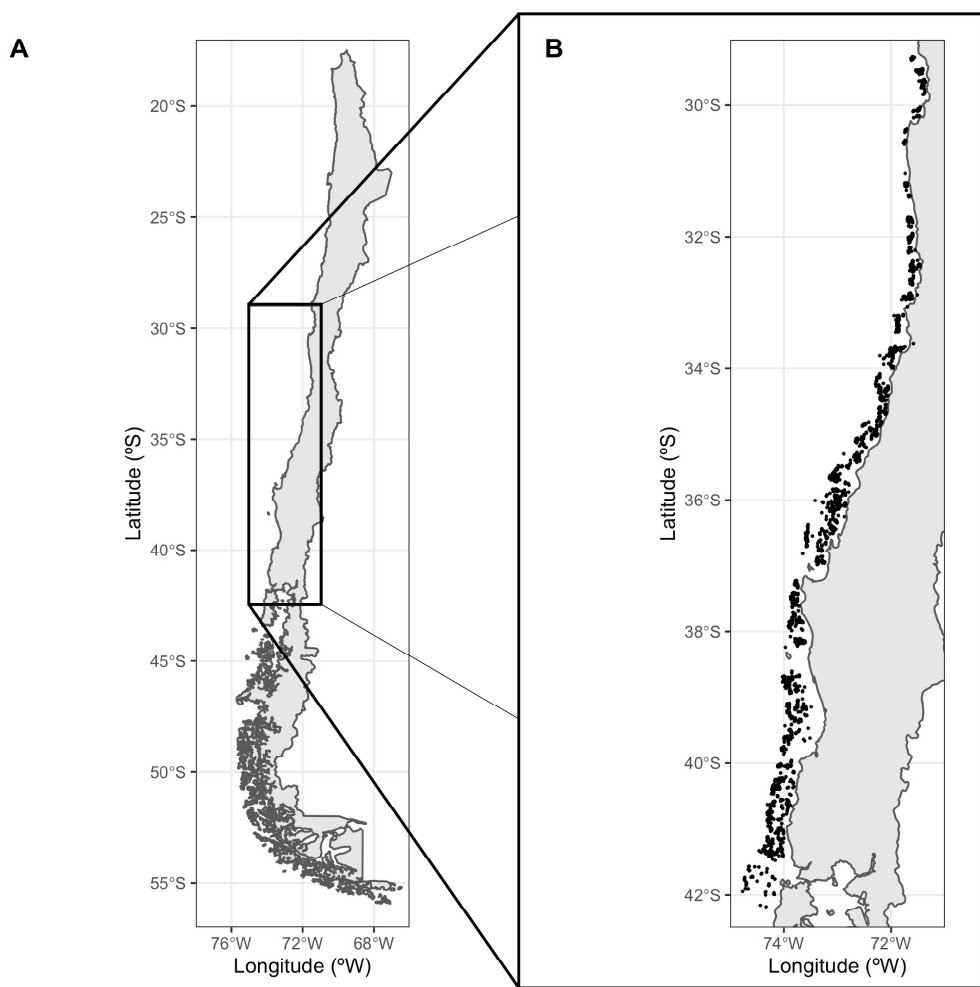

**Figure 1.** Study area shown in the square in Chile (**left**) and distribution of the fishing hauls (**right**) obtained during the Chilean hake stock assessment survey in the period between 1997 and 2018.

*2.2. Age Composition and Abundance per Unit Area*

Length–age keys were available by sex for each annual survey and obtained by sampling a fixed number of otoliths. The age composition for each fishing haul was obtained from length–frequency data according to the procedure summarized in Figure 2. Age–length keys (ALK) allowed us to obtain the probability that a fish of age *i* comes from

size $j$ ($q_{i,j}$); i.e., $q_{i,j} = a_{i,j} / \sum_j a_{i,j}$, where $a_{i,j}$ is the number of individuals with known age $i$ in length class $j$ [33]. For each year and fishing haul, we obtained the number of fish at length class $j$ ($f_{j,k}$) from length–frequency data per fishing haul ($k$). The length–frequency data were expanded to the catch (kg) of the fishing haul ($Y_k$) using the average weight, i.e., $C_{j,k} = (Y_k / \overline{W}_k) f_{j,k}$, where $C_{j,k}$ is number caught and $\overline{W}_k$ is the average weight (kg). Thus, the number of individuals caught by age results from multiplying the number of fish caught at size $j$ ($C_{j,k}$) by $q_{i,j}$, i.e., $A_{i,k} = q_{i,j} C_{j,k}$, where $A_{i,k}$ is the number of individuals at the age $i$ in set $k$. We obtained the number of fish at each age for each fishing haul, which ranged from age 0 to age 14+ (corresponding to fish that were 14 years old or older), and the swept area (km$^2$) allowed us to consider number per unit area (NPUA) as an index of abundance.

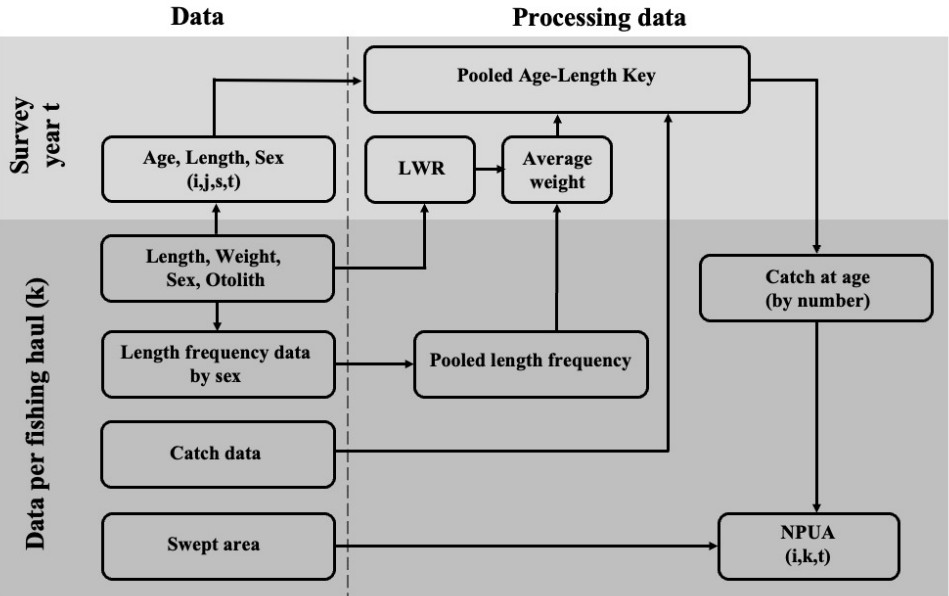

**Figure 2.** Survey data processing flow to obtain catch-at-age data, and hence an abundance index per age group i, fishing hauls k, and year t. Age (i), length (j), and sex (s) from fishing hauls allowed us to obtain a pooled age–length key (both sexes) for survey t, which was utilized to obtain catch at age from pooled length–frequency (both sexes). The abundance index is catch in number per unit of area, where the area is the swept area of each haul.

### 2.3. Spatial Distribution and Juvenile–Adult Relationships

We modeled the spatial distribution of Chilean hake juveniles considering the abundance of age groups 0 and 1 using generalized additive models (GAM), a flexible framework for modeling spatial and temporal effects and the relationship between co-variables through smoother functions [34]. Mainly, we used the "mgcv" package [35] for the statistical software R [36].

Abundance is represented by the NPUA by age groups, whose spatial and temporal distribution often involves a large proportion of zeroes in observations, i.e., zero-inflated data [16,20]. However, the proportion of zeros varied yearly, and the spatial pattern represents several annual realizations observed in each survey, and the sample size defined by the number of hauls was large enough (Table 1). Therefore, we chose a negative binomial distribution to represent the spatiotemporal variable, i.e., $Z(s,t) \sim \text{NB}(\mu(s,t), \theta)$, where $Z(s,t)$ represents the abundance in the spatial locations $s = 1, 2, \ldots, S$, and temporal index $t = 1, 2, \ldots, T$, and $\theta$ is the dispersion parameter. In GAM, we analyzed the expected number of fish on a log link according to the following expression:

$$\log(Z(s,t)) = \alpha + \sum_{i=1}^{I} f_i(X_i(s,t)) + \log(a(s,t)) \qquad (1)$$

where $\alpha$ is the intercept, $f_i()$ represents smoother applied to covariables $X_i(s,t)$, and $\log(a(s,t))$ is the logarithm of swept area, without a coefficient to represents the NPUA index; i.e., $\log(n_{i,t,k}) = \log(a_{i,t,k})$, where $n_{i,t,k}$ is the number caught of the age group $i$ in year $t$ and fishing haul $k$, and $a_{i,t,k}$ is the swept area. According to GAM nomenclature, we analyzed spatial and temporal effects separately on the abundance of hake juveniles according to the following linear predictor function:

$$n_i \sim s(t) + s(x,y,bs = \text{gp}, k = 200) + \text{offset}(\log(a)) \tag{2}$$

where $n_i$ is the abundance of either age group 0 or age group 1, and $s(t)$ is the temporal effects ($t$ = year) considering a smoother spline. $s(x,y,bs = \text{gp}, k = 200)$ is the smoother spline associated with spatial effects, i.e., the locations of fishing hauls where $x$ and $y$ represent longitude and latitude, respectively. The smoother considered a Gaussian process ($bs$ = gp), based on a Matérn covariance function [34,37], and we set k = 200 as an upper limit on the degree of freedom for smooths. Finally, the offset($\log(a)$) is the logarithm of swept areas ($a$) without estimating a coefficient. We omitted subindices for fishing haul and year for better representation of the model.

To investigate significant interactions, we chose a tensor product interaction to detect spatial, temporal, and spatiotemporal effects on the abundance of juveniles. Therefore, we added $ti(x,y,t,bs = gp, d = c(2,1), k = 200)$ to the previous model, and we evaluated significant interaction effects applying an analysis of variance (ANOVA) using a chi-squared test. In addition, San Martin et al. [28] found that juveniles tend to be in shallower waters. The abundance and presence, or detection probability, are often related [16,20,38]. Therefore, we added the bottom depth to complete the spatiotemporal distribution modeling of hake juveniles' abundance, i.e., $s(d, k = 10)$, where $d$ is bottom depth.

Once the spatiotemporal distribution model was completed, we analyzed the endogenous effects through the relationship between juvenile and adult abundance. The average age at maturity was approximately 3.5 years, but since 2004, it has been from around 2.5 to 3 years of age, and hake have begun to reach full maturity by age 5 [25]. Therefore, to avoid the transition from immature to mature individuals in the modeling, we utilized the NPUA of age groups 5, 6, and 7+. The latter is the sum of the abundance of seven-year and older adults. Accordingly, we incorporated the abundance of those age groups into the previous spatiotemporal model sequentially, applying a cubic spline smooth to NPUA of age 5 and 7+, e.g., $s(n_5, bs = cs)$, where $n_5$ is the NPUA of age group 5. In addition, considering that abundance of age 0 and 1 was correlated, we added the NPUA of age 1 to explain the abundance of age 0 juveniles by considering a yearly random effect smoother, i.e., $s(n_1, t, bs = re)$. Similarly, we incorporated the NPUA of age 2 for modeling juvenile abundance of age 1.

The nomenclature for labeling the model of the relative abundance of age group 0 and age group 1 was M0 and M1, respectively. Models M0.v and M1.v, where v = 1, 2, . . . , 7, represent models. We compared the models through the explained deviance and log-likelihood and utilized the Akaike Information Criterion (AIC) to select the best model, where $AIC = -2\log(L) + 2p$, $\log(L)$ is the maximum value of the likelihood function for the model and $p$ is the number of estimated parameters in the model [39]. In addition, we utilized the difference between AIC ($\Delta AIC$) and the relative weight of $\Delta AIC$ to compare the performance of the best model, i.e., $w_m = \exp(-0.5\Delta AIC_m) / \sum_{m=1}^{M} \exp(-0.5\Delta AIC_m)$, where $m$ is a given model from all models in the candidate set [40].

Finally, and with comparative purposes, we utilized GAM to describe changes in the abundance of the older adult hake (NPUE 7+). The structure of the model for older adult hake was like the spatiotemporal effect model for age 0 hake, including bottom depth.

## 3. Results

The abundance of the 0- and 1-year age groups of Chilean hake juveniles had similar patterns of variability, which expresses the positive correlation between these age groups

(Figure 3). Likewise, the abundance of juvenile age groups 0 and 1 negatively correlated with the abundance of adults of age groups 5, 6, and 7+. On the other hand, the abundance of adult Chilean hake showed positive correlations, particularly the abundance of age groups 5 and 6, and less with the abundance of older adults (7+).

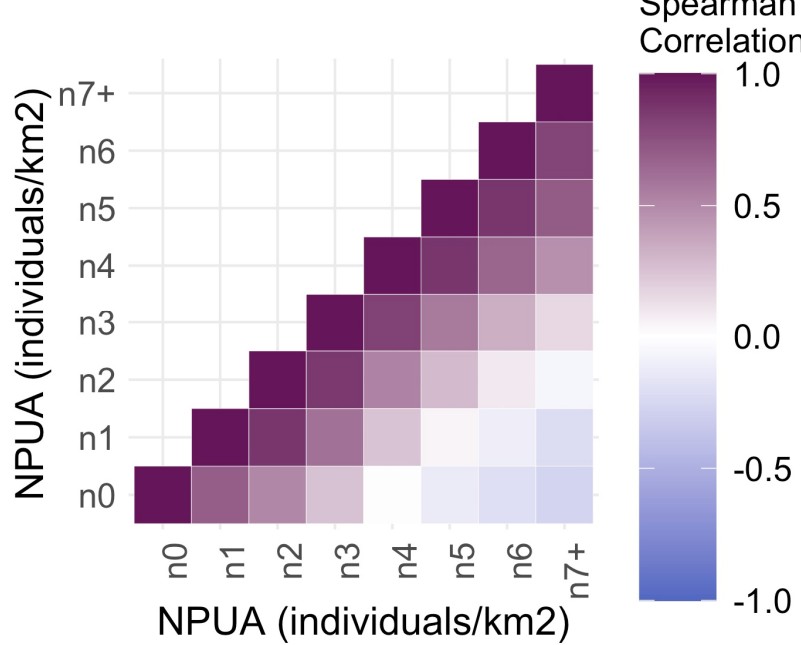

**Figure 3.** Coefficients of Spearman correlation obtained between the abundance index per age groups of Chilean hake, where n0, n1, . . . , n7+ represent the NPUA (individuals per km$^2$) of age groups 0, 1, 2, . . . , 7 and older.

The spatiotemporal interaction improved the performance of the M0 and M1 models (chi-squared test < 0.01), and the spatial distribution for the abundance of age groups 0 and 1 considering only spatiotemporal effects is summarized through models M0.1 and M1.1, respectively (Table 2). The models M0.1 and M1.1 explained 58.6 and 44.1% of the deviance, respectively. The dispersion parameters of the negative binomial showed low values in agreement with an excess of variability in observations and clumped distribution (Table 2). In addition, the bottom depth significantly improved the spatiotemporal models, increasing the previous deviance explained to 60.8% and 51.8% for models M0.2 and M1.2, respectively (Table 2).

**Table 2.** Models evaluated to explain the number of Chilean hake juveniles of age groups 0 ($n_0$) and 1 ($n_1$), during the austral winter, between 1997 and 2018. The GAM model considered the negative binomial distribution and link log. The number of age 7 ($n_7$) is the number per unit area of 7-year-old and older adult hake. The nomenclature is $t$ = year, $x$ = longitude, $y$ = latitude, $a$ = swept area, and $d$ = bottom depth. The GAM smother terms are explained in the text, and NB($\theta$) is the estimated dispersion parameter of negative binomial distribution. The selected model is M07 and M1.8 according to the Akaike's Information Criterion (AIC).

| Model | Linear Predictor | NB($\theta$) | Deviance Explained | Log-Likelihood | AIC | ΔAIC | AIC Weight |
|---|---|---|---|---|---|---|---|
| M0 | $n_0 \sim s(t) + s(x,y,bs=\text{gp},k=200) + \text{offset}(\log(a))$ | 0.138 | 55.5 | −9675.9 | 18,961.87 | 572.29 | 0.0 |
| M0.1 | $\text{M0} + ti(x,y,t,bs=\text{gp},d=c(2,1),k=200)$ | 0.147 | 58.6 | −9645.5 | 18,874.29 | 484.71 | 0.0 |
| M0.2 | $\text{M0.1} + s(d,k=10)$ | 0.160 | 60.8 | −9392.1 | 18,606.05 | 216.47 | 0.0 |
| M0.3 | $\text{M0.2} + s(n_5, bs=cs)$ | 0.163 | 61.5 | −9386.8 | 18,584.78 | 195.20 | 0.0 |
| M0.4 | $\text{M0.3} + s(n_1,t,bs=re)$ | 0.179 | 76.7 | −9333.4 | 18,420.24 | 30.66 | 0.0 |
| M0.5 | $\text{M0.2} + s(n_1,t,bs=re) + s(n_5,n_6)$ | 0.179 | 75.6 | −9327.5 | 18,418.53 | 28.95 | 0.0 |
| M0.6 | $\text{M0.2} + s(n_7,bs=cs)$ | 0.164 | 61.9 | −9381.0 | 18,571.10 | 181.52 | 0.0 |
| M0.7 | $\text{M0.6} + s(n_1,t,bs=re)$ | 0.181 | 75.9 | −9315.8 | 18,389.58 | 0.00 | 61.3 |
| M0.8 | $\text{M0.7} + s(n_5,n_6)$ | 0.182 | 76.8 | −9314.5 | 18,390.50 | 0.92 | 38.7 |
| M1 | $n_1 \sim s(t) + s(x,y,bs=\text{gp},k=200) + \text{offset}(\log(a))$ | 0.230 | 40.5 | −13,664.0 | 26,955.16 | 1333.84 | 0.0 |
| M1.1 | $\text{M1} + ti(x,y,t,bs=\text{gp},d=c(2,1),k=200)$ | 0.244 | 44.1 | −13,619.0 | 26,822.37 | 1201.05 | 0.0 |
| M1.2 | $\text{M1.1} + s(d,k=10)$ | 0.288 | 51.8 | −13,242.0 | 26,256.11 | 634.79 | 0.0 |
| M1.3 | $\text{M1.2} + s(n_5,bs=cs)$ | 0.295 | 52.9 | −13,227.0 | 26,198.38 | 577.06 | 0.0 |
| M1.4 | $\text{M1.3} + s(n_2,t,bs=re)$ | 0.365 | 95.1 | −12,936.0 | 25,641.25 | 19.93 | 0.0 |
| M1.5 | $\text{M1.2} + s(n_2,t,bs=re) + s(n_5,n_6)$ | 0.368 | 95.3 | −12,932.0 | 25,631.72 | 10.40 | 0.5 |
| M1.6 | $\text{M1.2} + s(n_7,bs=cs)$ | 0.289 | 52.0 | −13,240.0 | 26,251.62 | 630.30 | 0.0 |
| M1.7 | $\text{M1.6} + s(n_2,t,bs=re)$ | 0.365 | 95.1 | −12,938.0 | 25,644.33 | 23.01 | 0.0 |
| M1.8 | $\text{M1.7} + s(n_5,n_6)$ | 0.369 | 95.3 | −12,926.0 | 25,621.32 | 0.00 | 99.5 |

In addition, the endogenous component of the models also showed better explained deviance, especially the effects of the abundance of Chilean hake seven years and older (NPUA 7+) (Table 2). The best endogenous and spatiotemporal model for the abundance of age group 0 juveniles was model M0.7, in which the abundance of age group 1 was also included (Table 2). The deviance explained by model M0.7 was 75.9%. However, note that model M0.8 had a similar performance with an AIC weight of 38.7%. In addition, model M0.8 included the interaction between ages groups 5 and 6, which were correlated (Figure 3). Finally, the best model for the abundance of age one juveniles was model M1.8, including the abundance of age group 2 (Table 2). Model M1.8 explained 95.3% of the deviance with an AIC weight of 99.5%. Those models show that the spatial distribution of hake juveniles varied slightly across years between 1997 and 2018 (Figures 4 and 5).

In these models, and according to partial effects, slight differences in temporal, spatial, and depth effects and negative effects of the abundance of older adults were observed. The abundance of juveniles at age 0 tended to be distributed along the coast with relative maxima between 35 °S and 40°30′ S, peaking at 29 °S, 36 °S a 37°30′ S (Figure 6A). The western distribution tended to decline offshore after 73°30′ W (Figure 6B). The abundance of juveniles of age 0 was located over the continental shelf (<200 m), declining faster after the shelf break (>300 m) (Figure 6D). The temporal effect showed an increment in the abundance from 2002 to 2007, subsequently remaining at high levels (Figure 6E). The endogenous effects show the negative influence of 7-year-old and older adult Chilean hake abundance, and positive effects of age-1 juveniles (Figure 6E,F).

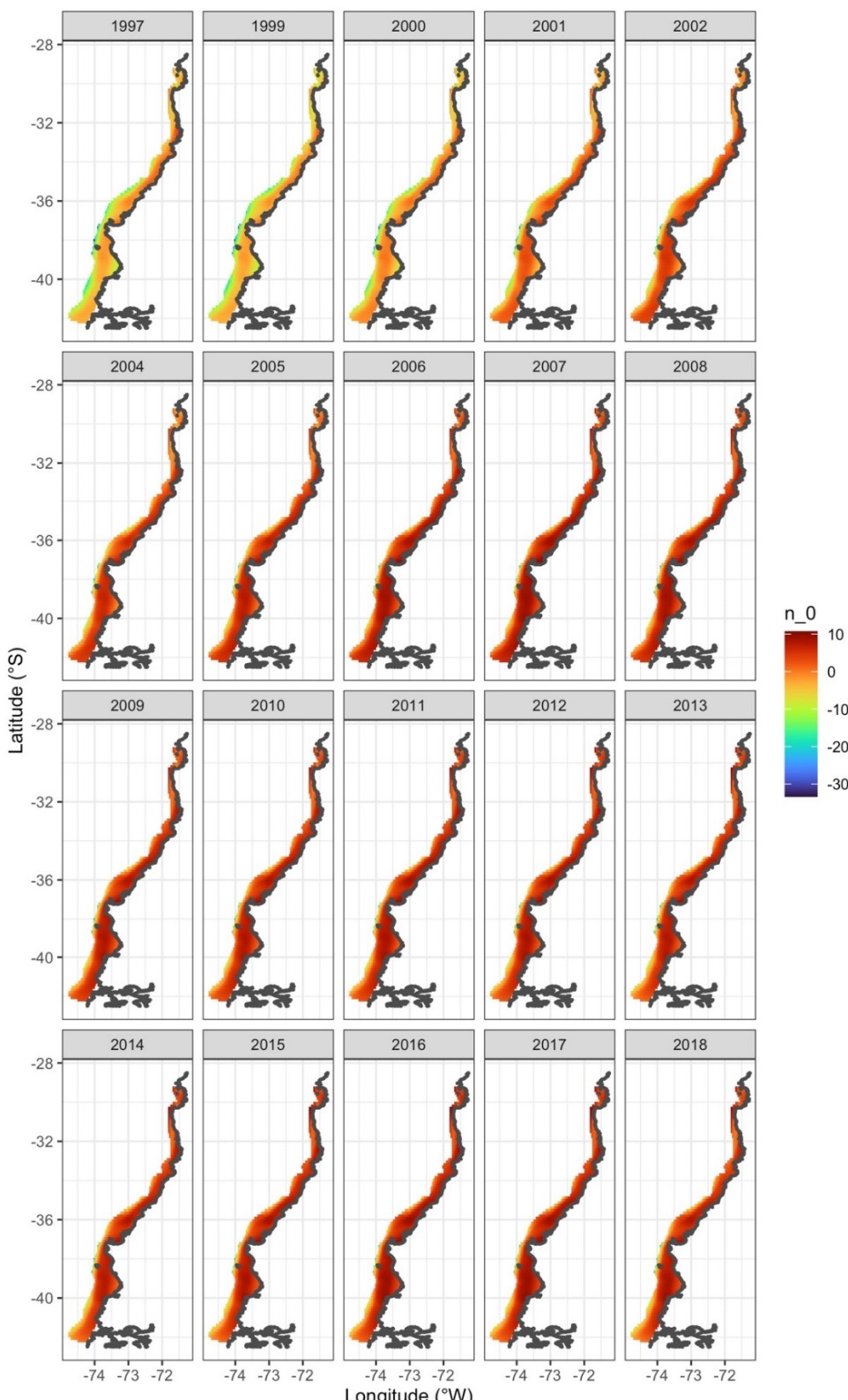

**Figure 4.** Spatiotemporal distribution of the abundance of Chilean hake juvenile of age 0, from 1997 to 2018 (Model M0.7, Table 2).

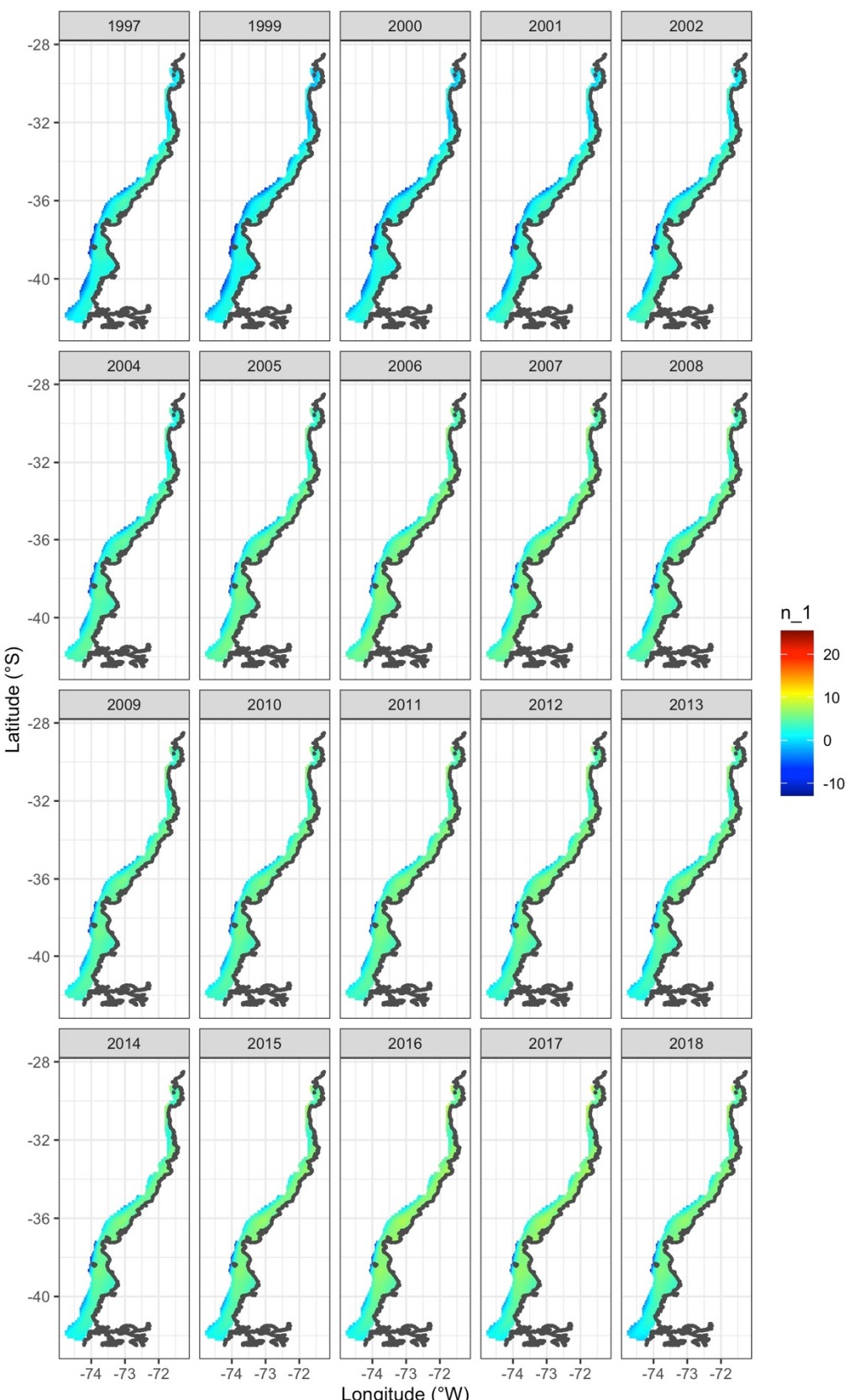

**Figure 5.** Spatiotemporal distribution of the abundance of Chilean hake juvenile of age 1, from 1997 to 2018 (Model M1.8, Table 2).

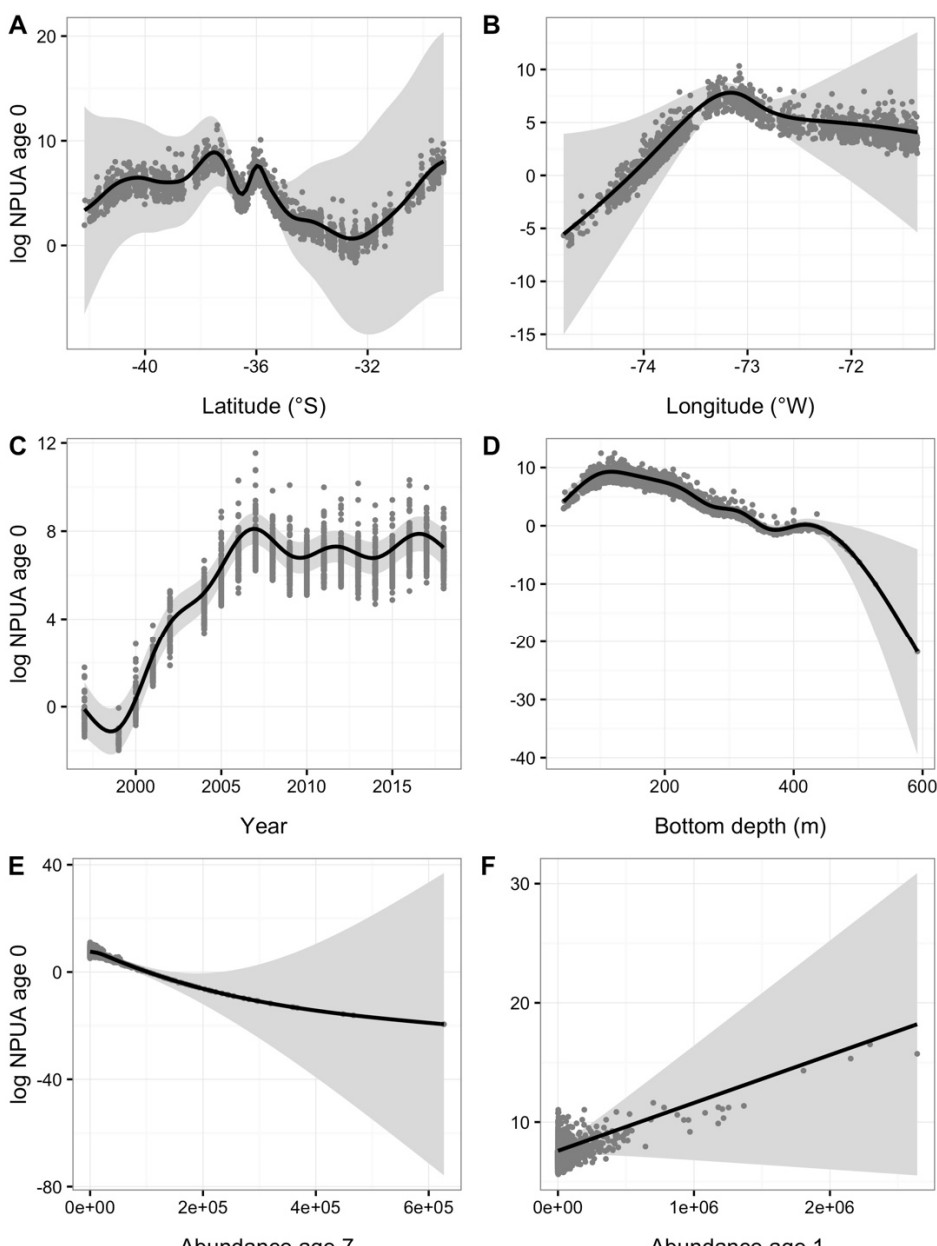

**Figure 6.** Relationships between the abundance of juvenile Chilean hake of age 0 and relevant variables for spatiotemporal effects latitude (**A**), longitude (**B**), year (**C**), bottom depth (**D**) and endogenous effects of older adult hake of age 7+ (**E**) and juvenile hake of age-1 (**F**), according to the best model fitted (Model M0.7, Table 2).

The abundance of age one increased between 36 °S and 40 °S, increasing toward the north of 32 °S (Figure 7A), tended to decline offshore, from 73 °W to 75 °W (Figure 7B), and was located over the continental shelf (<200 m), declining after the shelf break (Figure 7D). The temporal effects showed an increment in abundance after 2000, stabilizing with fluctuations after 2007 and peaking in 2015 (Figure 7C). The endogenous component of the model showed negative effects of older adult hake (7+) (Figure 7E) and positive effects of 2-year-old juvenile hake (Figure 7F).

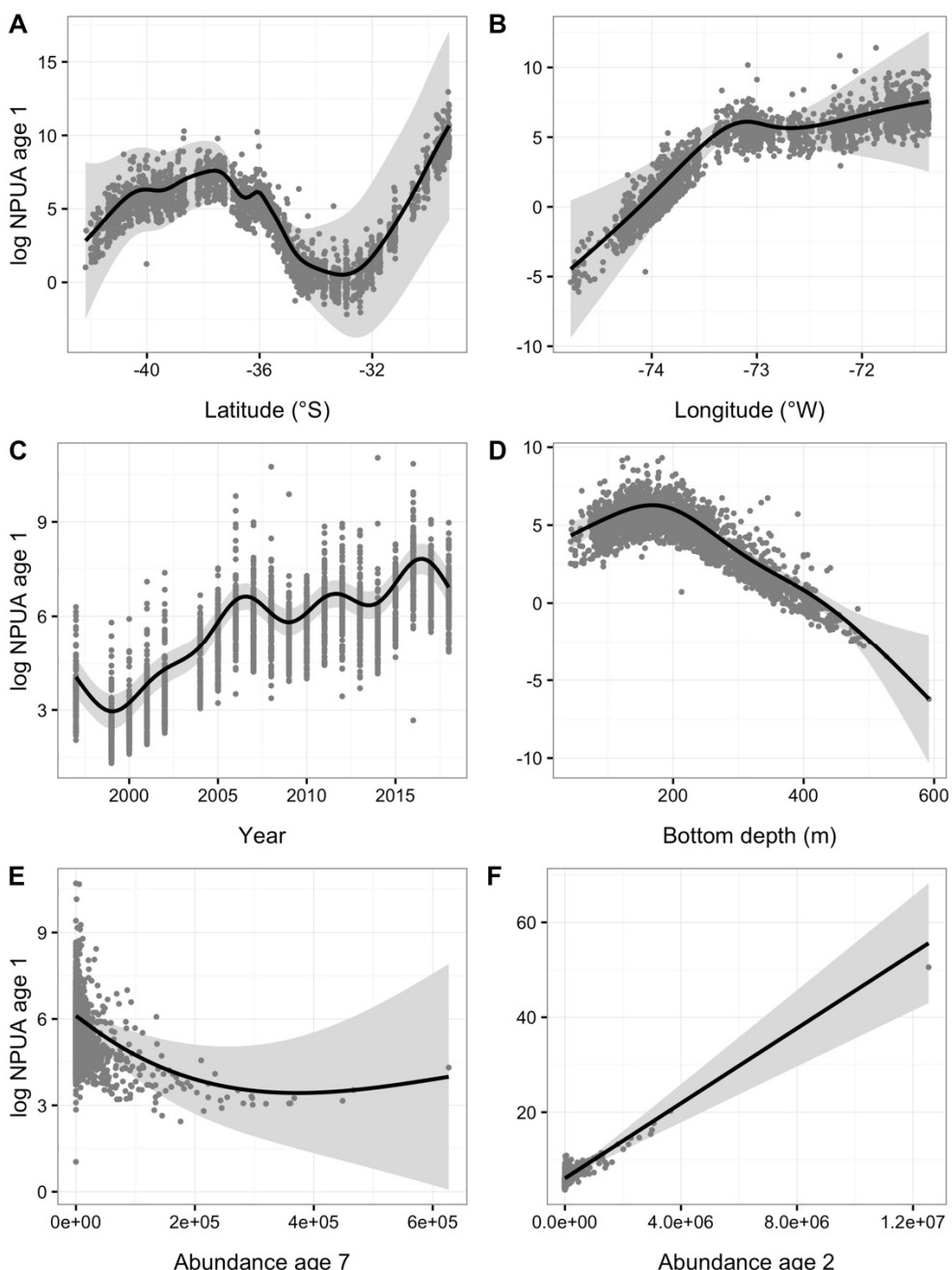

**Figure 7.** Relationships between the abundance of juvenile Chilean hake of age 1 and relevant variables for spatiotemporal effects of latitude (**A**), longitude (**B**), year (**C**), bottom depth (**D**), and endogenous effects of older adult hake of age 7+ (**E**) and juvenile hake of age-2 (**F**), according to the best model fitted (Model M1.8, Table 2).

With a comparative purpose, the spatiotemporal model for the abundance of seven-year-old and older hake was significant ($p < 0.01$), with an explained deviance of 40%. Thus, the abundance of the younger and older fraction of the stock fluctuated in opposite trends, and the abundance of seven-year-old and older adult hake tended to be distributed at a deeper bottom depth (>200 m) than juveniles of age 0 (Figure 8). Note that there were few data for the depths deeper than 500 m, determining an increase in the confidence limits (Figure 8B,D).

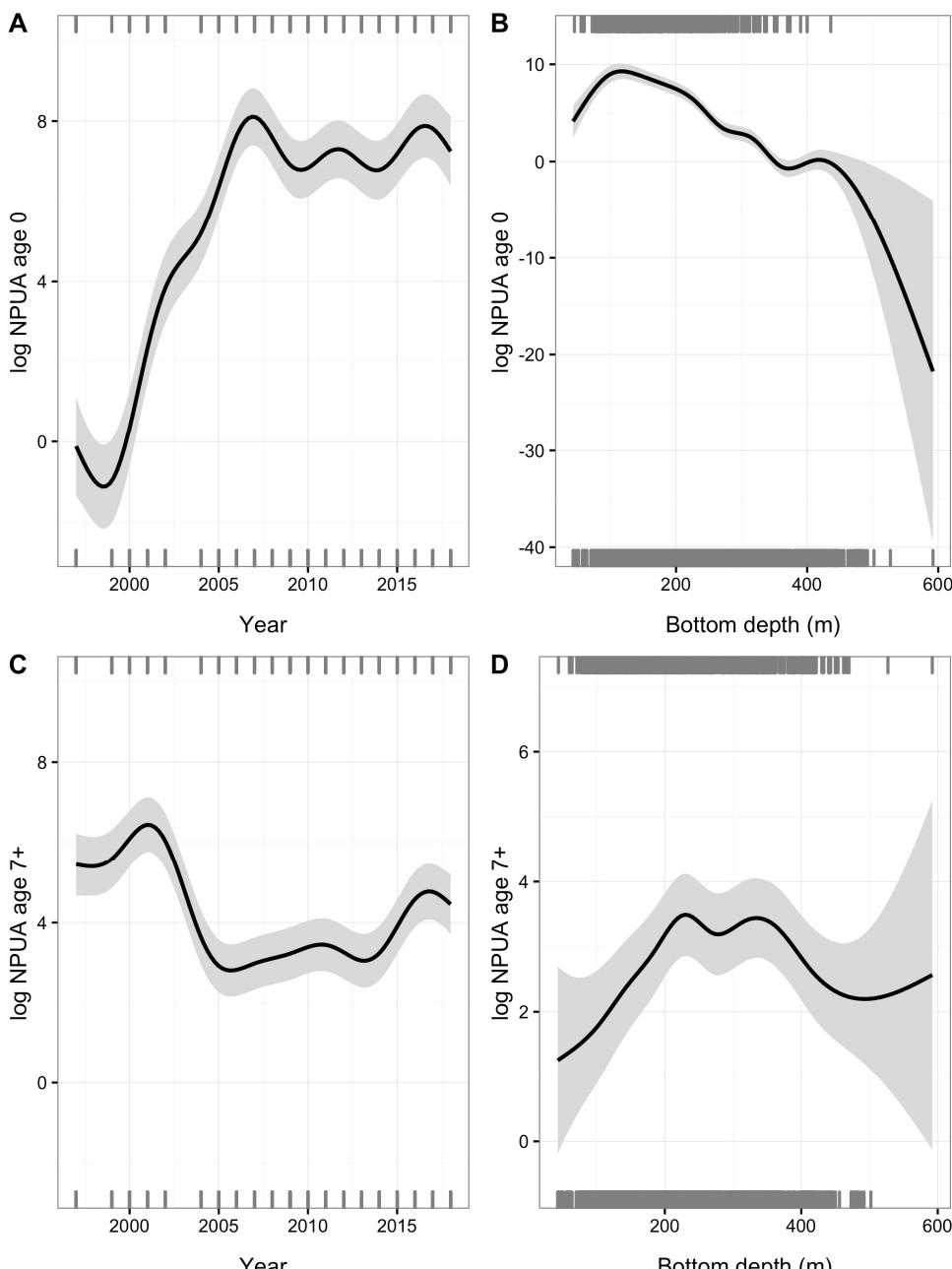

**Figure 8.** Temporal changes in log abundance and distribution of the abundance as a function of the bottom depth: log abundance of age 0 (**A**,**B**), and log abundance of age 7+ (**C**,**D**).

## 4. Discussion

The relationship between youth and adults in a space–time context indicates population processes associated with endogenous effects on abundance. However, if the juvenile spatial pattern is due to negative interactions with adults, the juvenile abundance may move across the area through time, i.e., changing its spatial distribution [16]. In hake, the spatial distribution of juveniles changed yearly due to changes in abundance rather than in spatial structure. Indeed, the temporal effects showed that the abundance of juvenile hake grew from 2002 to 2007, reaching subsequent stability at higher levels. Thus, juveniles of age 0 and 1 showed similar spatial patterns in abundance, particularly between 35 °S and 40°30′ S and north of 32 °W in shallower depths over the continental shelf, and declining abundance at depths lower than the shelf break. The latitudinal pattern in the younger abundance of hake is coincident with the offshore extension of the continental shelf, which

tends to be wide, particularly from 35 °S to 37 °S and from 38 °S to 40 °S [41]. In these areas, the continental shelf and shelf break are bathed by the subsurface Peru–Chile Current flowing towards the pole [42], and the mixing of water masses between the Equatorial Subsurface Water ($O_2 < 1$ mL $L^{-1}$, $12-13$ °C, 35 psu). and the Antarctic Intermediate Water (oxygenated and cold water, $11-12$ °C) [43]. Thus, the preference of juvenile hake for shallower waters could be associated with oxygenated waters, but environmental variables contribute little to explain the presence–absence of abundance of Chilean hake [28,29].

Specimens of age 2 were immature individuals before 2004 [25]. However, after 2004, fish of two years reached a proportion of 48% maturity due to the reduction in the maturity length [25,44]. Therefore, immature juveniles of age one began to move towards deeper waters as they grow older [28], explaining the positive correlation of ages 0, 1, and 2, and the negative correlation from age five onwards. Other species of the Merluccius genus show that juveniles prefer to occupy shallow areas [18,45,46], and exhibit an ontogenetic migration from the nurseries to the end of the continental shelf [47–49]. Therefore, to minimize the probability of harvesting juvenile age groups during the recruitment process, it is essential to restrict bottom-trawling operations to areas deep enough to consolidate the recruitment process. In Chile, the first five nautical miles of the coast are exclusive to small-scale fishing operations, and industrial bottom trawling is not allowed. In this context, to avoid fishing for age-0 and -1 juvenile recruiting, it is essential to ensure an optimal selection of artisanal spinel and gillnet fishing gear [50,51].

Previous studies showed that juveniles' presence started in 2004 [22,28]. However, these studies designated fish smaller than 34 cm in length as juveniles. Here, we used abundance by age to study endogenous effects on age- 0 and -1 fish. We know that fish abundance can vary in response to environmental variables [9,52–54], migration [18,55,56], and fishing pressure [57,58]. Hake seems to tolerate different environmental conditions due to the wide range of habitats it occupies, from benthic to pelagic through diel migrations [24,28,29,59]. In this context, exogenous factors would not be limiting the distribution of 0- and 1-year-old juveniles, except in the bottom depth and the spatial configuration of the coast, in terms of extension and reduction in the continental shelf and shelf break along the coast. We conclude that the expansion in the abundance of juveniles after 2002 and consolidations after 2007 was favored by the low abundance of older adult hake. However, is important to study the overlap between juveniles and adults in a spatiotemporal context using more rigorous spatiotemporal analysis, such as Bayesian hierarchical models, to identify refuges or nursery areas along the coast [38]. Indeed, generalized additive models (GAM) could only smooth the spatial trend, and autocorrelation in residuals could be approximated by mixed GAM (GAMM) [34]. In part, we used a Gaussian field approach for the smoother spatial term through the covariance function suggested by Kammann and Wand (see details in [35]). Still, the hotspot of juveniles could be identified by hurdle models or by calculating the exceedance probabilities of juvenile abundance being more significant than a given threshold value that is of interest for managers and fishers.

Age-0 and -1 fish could seek refuge in neritic habitats, which would allow them to increase survival by decreasing the risk of predation [31,60], particularly the predation of hake by their ichthyophagous congeners [32]. The negative relationship between the abundance of juveniles at ages 0 and 1 and older adults (7+), shows the potential pressure cannibalism could indirectly exert on younger ages [4,30,31]. These relationships suggest that removing adults by fishing (overexploitation, illegal capture, and discarding) [24,26] could expand the spatial distribution of juveniles in the stock, such as the higher occurrence of juveniles after 2003. These results could contribute to fisheries' management, recognizing that the juvenile expansion in space–time would help the stock's sustainability over time. Therefore, it is essential to identify juveniles' nurseries or recurrent areas outside the fishing grounds and quantify their contribution to the adult stock. The results here are a good starting step to understanding the fundamental factors of the Chilean hake population ecology that are currently not accounted for in stock assessment and management.



**Author Contributions:** Conceptualization, D.V.Y. and L.A.C.; methodology, L.A.C.; formal analysis, D.V.Y. and L.A.C.; investigation, D.V.Y.; resources, L.A.C.; data curation, D.V.Y.; writing—original draft preparation, D.V.Y.; writing—review and editing, L.A.C. and H.A.; supervision, L.A.C. and H.A.; project administration, L.A.C.; funding acquisition, L.A.C. All authors have read and agreed to the published version of the manuscript.

**Funding:** This study was funded by Agencia Nacional de Investigación y Desarrollo (ANID) through scholarship ANID-PFCHA/Doctorate National/2017-21170986 to DVY, and LAC was partially funded by Grant COPAS Sur-Austral (ANID PIA APOYO CCTE AFB170006) and COPAS COASTAL (ANID FB210021).

**Data Availability Statement:** The data that support the findings of this study are available from the corresponding author upon reasonable request.

**Acknowledgments:** D.V.Y. and L.A.C. thank the "Fondo de Investigación Pesquera y acuicultura" https://www.subpesca.cl/fipa (accessed on 7 April 2022) and Instituto de Fomento Pesquero https://www.ifop.cl (accessed on 7 April 2022) for facilitating the final reports and data associated with the acoustic surveys of Chilean hake.

**Conflicts of Interest:** The authors declare no conflict of interest.

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
