# Peer review of "Juvenile Hake Merluccius gayi Spatiotemporal Expansion and Adult-Juvenile Relationships in Chile"

_fishes, doi:10.3390/fishes7020088_

Round 1

Reviewer 1 Report

Dear Author,

I have a few concerns about your manuscript. In my viewpoint, you plotted some figures in the Discussion part of your article.It should be your results. Would be fine if you can adjust those figures in your results. Indeed, it will enhance the quality of the article.Also, you can revise the English of the manuscript by a native.

Author Response

Thank you very much for your suggestion. Figure 8 was in the discussion section and moved to the results section in the revised version of the manuscript. In addition, a native speaker reviews the English grammar. 

Reviewer 2 Report

Yepsen et al. present an interesting study on spatiotemporal distribution and juvenile-adult relationships in Chile's continental shelf and shelf breaks areas. Data was obtained from bottom trawls and the study represents solid time series. They conclude that juvenile expansion after 2002 happened as a result of the decreased abundance of older hake, which preyÅ› on juveniles hake.

I recommend the study for publication after the minor revision. Please find my comments from pdf file as text margins.

Author Response

Thank you for your review and suggestions. We modified the text according to your minor requests and incorporated it into the revised version of the manuscript in red text.

1) Line 2, Would recommend despite of descriptive tittle, name the main results "Juvenile hake Merluccius gayi expansión in Chile was governed by…". 

R.: Accordingly, we changed the title of the manuscript to "Juvenile hake Merluccius gayi spatiotemporal expansion and juvenile-adult relationships in Chile.

2) Line 18: What kind of negative effect? Please specify

R.: Specified in Line 18 "...on abundance of young generation".

3) Line 18, Is that a result of trend analyses, or what are the independent factors studied? Please specify.

R.: Specified in Line 18-19

4) Line 25, Better elaboration, in terms of how this study complements existing knowledge and spatiotemporal distribution improves hake stock management, is needed.

R.: "Our results highlight the importance of endogenous factors in the spatial distribution of Chilean hake juveniles to identify nurseries or juvenile areas free of potential cannibal adults." (Line 24).

5) Line 26, Hard to read, recommend rewording: It is critical to understand causes of commercially exploited population distribution.

R.: Done.

6) Line 57, what about recruitment effect – it should affect age composition of the stock?

R.: We agree, weak and robust recruitment affects the age composition, but here we are examining causes that could enhance recruitment. (Line 58)

7) Line 71, change excepting to except.

R.: Done.

8) Line 72, Better elaboration of data utilized in this study is needed to understand and evaluate how comparative they are between years.

R.: The hake surveys were executed considering the same methodology, same research vessel, and almost the same researcher as documented in the cited reports—the grant number allocated to each survey help to look for a specific year and data.

9) Line 80, Since this paper is about the distribution of hake by age, it would be important for reader to provide a figura showing empirical evidence of age-length relationship with variation. In other Word figure showing in what extent age varies by length.

R.: I think we can prepare a figure like that and document it as supplementary results. However, we want not to deviate the attention because the empirical age-key can be consulted in each survey report, where they are fully communicated.

10) Line 88, Figure 1 There is two maps. Please explain in figure caption? What is denoted with dots, what in square?

R.: We improved Figure 1 showing the study area (See Fig. 1).

11) Line 91, on survey code, This one is confusing. Until 2013 it seems to be year+month/nr of trawls? After that, it is hard to understand what it means? 

R.: The code is the grant number of each survey. We improve this in the text and legend of Table 1. The Chilean Research Fund assigns the survey code according to its internal administrative rules. Until 2013, the cruise was funded by the "Fondo de Investigación Pesquera y Acuicultura", and from 2014 the team in charge was IFOP, changing the codes.

12) Line 104, Please explain in the methods how you gained swept área.

R.: The swept area is part of the database and reported. (See line 79).

13) Line 138, I don’t have sufficient statistical background to evaluate of statistical analyses.

R.: The methods are the standard for this kind of models.

14) Line 182, This confused me, as normally such correlation is made between age x and age x+1. The rationale is to see whether they follow similar dynamics or variability. But, how should different ages in the same year correspond to similar dynamics/variability as the number of individuals depends on year-class strength and fishery?

R.: The rationale for including the age x+1 is because age 0 must grow into age 1 close to the same areas in a given year. Because they are correlated, it is better to include them as a random effect rather than an independent effect. However, there was a typing error, and we corrected it (line 184). Now only the correlation between age x and age x+1 is mentioned.

15) Line 230, Figure 4. Why you built the map base don model M0.1 and not M0.7 that was the best model?

R.: The idea was to compare age 0 and age 1 under the same distribution. However, we replace the figures according to the best models. 

16) Line 234, Figure 5. Why not M1.8 that explained the most of the variance

R.: Idem (see the previous answer).

17) Line 260, In general: discusión could be expanded. Important questions to answer: 1) how these results help to improve the management of hake stock? 2) what are the environmental variables shaping the distribution patterns of hake?

R.: We are clarifying in this research that a predation effect would drive these movements. In this case, when the fishery removes a predator (old ages), it would favor the distribution of juveniles by widening their range. According to the previous literature, environmental variables have minor effects on the distribution (see San Martin et al. 2011; 2013). (see line 300-301). We expand the discussion by emphasizing management issues at the end of the paper (Lines 344-350).

18) Line 267, delete “in abundance”

R.: Done (see line 286).

19) Line 280, delete “The recruitment to the exploitable stock is at age 2”.

R.: Done (see line 302).

20) Figure 6, How the abundance can get below zero? Although, I anticípate its a modeled predicted outcome, I believe better to use Poison or other scale to be les confusing.

R.: In GAM, the effects on the dependent variable is based on the link function. In this case, is "log", reason why the effects on the dependent variable are plotted for each covariable on the scale of the linear predictor. 

21) Figure 7, E: Trend since this point (above 4e+05 is driven by too few variables. F: I recommend not to display this single point at the highest level, which drives this trend.

R.: To let the data talk by themselves, we do not discriminate in the results. However, the influence of high abundance indeed has a negative partial effect on age 1 on the spatial field. The negative trend in time of the average abundance is shown in Figure 8. 

22) Figure 8, B: Bottom Depth above 500m seems to be covered with only very few observations – how to reliable this results is, as there seems to be Sharp drop in abundances? Should be discussed accordingly in the text.

R.: Yes, Few data on deeper than 500 m because most of the survey is carried on the continental shelf (< 200 m) and shelf break (< 500 m). (se line 275-276). 

Reviewer 3 Report

This is a good paper, but I have a big question that involve all the analysis.

Is the GAM a good method for spatial statistics?

My answer is NOT.  When you put in a model a spline over the surface x,y you only are considering that exist any relation, but you don't are considering de spatial correlation, in  frequentist o Bayesian approach that are several techniques  that considerer the spatial correlation, as a real spatial interaction, not the longitude-latitude possition.

Are you compared the results width one of these techniques? 

Author Response

Is the GAM a good method for spatial statistics?
My answer is NOT. When you put in a model a spline over the surface x,y you only are considering that exist any relation, but you don't are considering de spatial correlation, in frequentist o Bayesian approach that are several techniques that considerer the spatial correlation, as a real spatial interaction, not the longitude-latitude possition.
Are you compared the results width one of these techniques? 

R.: Thank you for your observation. GAM can be a suitable method for identifying spatial trends on a large scale, such as the data for hake in our paper. Of course, the spatial correlation among neighboring must be considered on a small scale, where the emphasis is on the aggregations that can be part of the same patch. In this case, GAM models are not good. However, the discussion issue is beyond our paper. We are trying to understand the importance of partial endogenous effects of adults on juvenile expansion. Since we added a spatiotemporal interaction term in GAM and physical covariables (bottom depth), the partial effects of juvenile-adult relationships are in the presence of the spatiotemporal interaction and bottom depth results. We changed the title of the manuscript, and we discussed part of the approach in lines 329-336.
We used a Gaussian process (bs=gp), based on a Matérn covariance function, but we did not compare the technique. Furthermore, our emphasis was not on identifying spatial correlation. Instead, we think of identifying small areas and juvenile hotspots (nurseries areas) with additional methods such as Bayesian spatiotemporal hierarchical models. 

Round 2

Reviewer 3 Report

Thanks for attending my suggestions. This new version improve the previous one.